



# Visual Interpretation of Synthetic Aperture Radar Sea Ice Imagery by Expert and Novice Analysts: An Eye Tracking Study.

Alexandru Gegiuc[1], Juha Karvonen[1], Jouni Vainio[2], Eero Rinne[1,3], Roman Bednarik[4], and
Marko Mäkynen[1]

[1]Finnish Meteorological Institute(FMI), Marine Research, Erik Palménin aukio 1, 00560 Helsinki, Finland
[2]Finnish Meteorological Institute(FMI), Ice Service, Erik Palménin aukio 1, 00560 Helsinki, Finland
[3]University Centre in Svalbard (UNIS), Arctic Geophysics, Longyearbyen, Norway
[4]University of Eastern Finland (UEF), Joensuu, Finland

**Correspondence:** Alexandru Gegiuc (alexandru.gegiuc@gmail.com)



**Abstract.**

We demonstrate the use of eye tracking methodology as a non-invasive way to identify elements behind uncertainties typically introduced during the process of sea ice charting using satellite synthetic aperture radar (SAR) imagery. To our knowledge, this is the first time eye tracking is used to study the interpretation of satellite SAR images over sea ice. We describe differences and similarities between expert and novice analysts while visually interpreting a set of SAR sea ice images.

In ice charting, SAR imagery serves as the base layer for mapping the sea ice conditions. Linking the backscatter signatures in the SAR imagery and the actual sea ice parameters is a complex task which requires highly trained experts. Mapping of sea ice types and parameters in the SAR imagery is therefore subject to an analyst's performance which may lead to inconsistencies between the ice charts. By measuring the fixation duration over different sea ice types we can identify the features in a SAR image that require more cognitive effort in classification, and thus are more prone to miss-classification. Ambiguities in classification were found especially for regions less restrictive for navigation, consisting of mixed sea ice properties and uneven thicknesses. We also show that the experts are able to correctly map large sea ice covered areas only by looking at the SAR images. Based on the eye movement data, ice categories with most of the surface covered by ice, i.e. in ice charts fast ice and very close ice, were easier to classify than areas with mixed ice thicknesses such as open ice or very open ice.

# 1   Introduction

Maritime shipping in cold regions requires up-to-date information about sea ice conditions, typically over large areas. This information is usually provided by experts trained for analyzing sea ice. They rely on satellite imagery and in-situ observations to describe the ice conditions by means of ice types and parameters. In the Baltic Sea, daily ice charts produced during winter by the Finnish Ice Service (FIS) contain manually drawn polygons of distinct ice types (WMO-JCOMM, 2014) and related parameters, such as sea ice concentration, thickness or degree of ridging, indicated by symbols or values on the ice chart. These charts are mainly based on the visual interpretation of synthetic aperture radar (SAR) imagery and messages from ice-breakers. Visual discrimination of sea ice features in SAR imagery is a complex task because the SAR radar signature is mainly dependent on sea ice surface roughness. Thus, additional knowledge on the regional and local weather, topography and sea ice conditions are required by the analysts to be able to translate the SAR information into relevant geophysical ice characteristics. Furthermore a temporal gap in the SAR or additional sea ice data available to analysts results in increased uncertainty (Gegiuc et al., 2018).

We used an eye tracker to record and compute the gaze points on the screen from each analyst, while looking at, identifying and classifying sea ice features in five SAR images acquired over the Baltic Sea. Our study is divided in two parts. First we investigate the interaction between both novice and expert analysts and SAR imagery during visual identification and classification of sea ice features. Here we pay attention to at the time the analysts use for analyzing the images and the sea ice types or features identified, interaction with the SAR images such as zooming or panning the image content and other differences or similarities between experts and novices.



In the second part we explore the relationship between gaze data of experts analyzing SAR features and the actual sea ice classification results. We were interested in identifying individual differences between experts in their classification results and how these are linked with their eye movements. We also compare the eye tracking fixation data to the automated analysis of the SAR image local complexity.

5 ## 1.1 Sea ice charting in the Baltic Sea

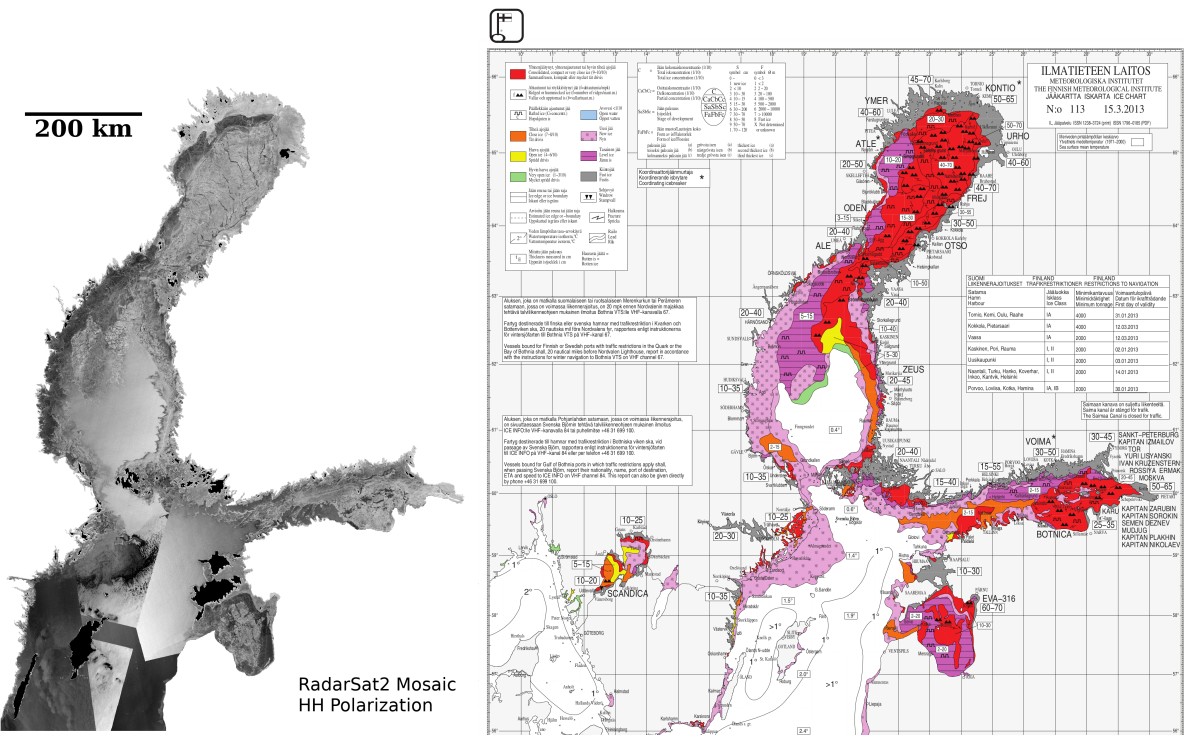

**Figure 1.** An example of SAR mosaic with 500 m pixel size over Baltic Sea (Bay of Bothnia), processed from RADARSAT-2 SAR HH-polarization imagery (left), and the corresponding Finnish Ice Chart (right) (© Finnish Meteorological Institute) from 15th of March 2013.

The basis for the Baltic Sea ice charting at FIS is daily SAR mosaic which is constructed automatically by merging together independent SAR images. The mosaic is updated once per day, typically in the morning, to include most recent available SAR images. Figure 1 shows an example of the daily SAR mosaics and the corresponding sea ice chart manually drawn by the FIS analysts.

10 When constructing an ice chart, historical information is always used as the base layer for the analysis. Using the most recent data, experts classify SAR mosaic regions into several sea ice types as defined in (WMO-JCOMM, 2014). The new chart consists of polygon boundaries and the ice type within each ice polygon based on the available information, most of which comes from a fresh SAR frame. Due to time constraints, FIS analysts typically draw large polygons i.e. tens of kilometers while suppressing small scale features such as *leads* or *cracks*. Parameters such as the degree of ridging in a qualitative scale is





then added on top of each polygon as symbols. Similarly, sea ice thickness is added to the chart as average level ice thickness for each polygon.

The visual interpretation of a sea ice SAR frame is known to be a complex task, as it requires extensive knowledge on the typical sea ice dynamics, current and historical weather and ice conditions as well as good understanding of the relation

between the radar backscatter signature and various sea ice types in different weather conditions. Subsequently, analysts have to combine their information from SAR imagery with other available information and apply their own expertise to be able to correctly classify a sea ice region. A detailed description of the ice charting practices can be found in the ice charting manual (MANICE, 2005).

For the Baltic Sea, ice concentration, mean level-ice thickness and ridge density are the three main parameters reported in the

FIS ice charts. The colors in the ice charts can be regarded as semaphore lights for shipping, distinguishing between different ice types (see Table 1). Without ice-breakers' assistance the red color represents "stop" or "no go" and consists of *consolidated* or *very close ice*. With ice-breakers' assistance navigation in these areas is possible. *Close ice* marked with orange color may be restrictive (difficult) for weaker ice class ships and it often contains mixed ice types difficult to predict. Yellow color marks regions of *open ice* where weak ice class ships should "proceed with care". Green and other colors are all considered navigable

areas with no restrictions. Consolidated land *fast ice* is marked with gray color.

**Table 1.** Ice Chart - traffic color lights with the corresponding recommendation per ice class.

| Ice class | IC (%) | Recommendation |
|---|---|---|
| *Very close ice* | 9-10/10 | No Go, Stop. IB OK |
| Close ice | 7-8/10 | Proceed with IB guidance, IB OK |
| Open ice | 4-6/10 | Proceed with care, IB OK |
| *Very open ice* | 1-3/10 | Go, OK |
| *Open water* | < 1 | Go, OK |
| *New ice* | 71% | Go, OK |
| *Level ice* | 91% | Go, OK |
| *Fast ice* | 100% | No Go, Stop. IB OK |

*I*C = ice concentration; IB = ice-breaker.

Automated classification methods of sea ice parameters from SAR imagery exist as well, see an overview in Zakhvatkina et al. (2019).They typically employ a similar approach as the manual ice charting. First pixels with similar texture features, intensities, patterns are grouped into segments which are statistically analyzed and classified based on their correlation degree with known ice classes or parameters. The classification decision can be based on training data, or on reference data (e.g.

Karvonen et al. (2015)). One example of the automated Baltic Sea ice products based on the SAR imagery is recently developed degree of ice ridging product in Gegiuc et al. (2018). There a direct comparison with the manual ice chart is also shown and discussed in more detail.



Even though both manual and automated charts aim to have high accuracy, consistency and reliability, significant differences between the two products have been reported in literature (Agnew and Howell (2011),CIS (2006),Gegiuc et al. (2018),Karvonen et al. (2015)). Karvonen et al. (2015) presents a comparison between the ice concentrations estimated manually and automatically. The automatic analysis is aimed to support the ice charting work and perhaps replace the manual estimates in 5 the future. In their study, differences in ice concentration estimates between expert groups were found, together with differences between manual and automated estimates. This underlines that uncertainties and inconsistencies in the ice charting can be introduced by experts doing the ice analysis. However, there are currently no means of measuring this kind of uncertainty introduced by the analysts in the ice charts.

## 1.2 Visual interpretation of imagery and digital images

10 Human visual interpretation of scenes has been studied for long in many domains, including aviation (Wickens et al., 2005), medicine (Ahmidi MS et al., 2012; Eivazi et al., 2012; Erridge, 2018) or human behaviour studies (Gordon et al., 2007). In aviation, for example, errors in plane manoeuvres that lead to fatalities are mostly linked with errors in situation assessment, and more specifically errors related to attention allocation of the pilots (Wickens et al., 2005).

Visual interpretation of complex scenery plays an important role in medicine, where for example neurosurgeons perform 15 critical decision-making tasks during surgery procedures. In those cases, even the smallest error can be fatal to patients (Eivazi et al., 2012; Erridge, 2018). In image-based diagnosis or any other medical diagnosis which is based on the visual interpretation of a medical scan, the quality and speed of visual information processing are crucial with a direct impact on people's health. Consequently, by measuring the level of expertise of analysts, their focus or cognitive effort, can help increase safety or minimize errors in the performed visual recognition tasks.

20 A typical fixation duration of a human observer is around 0.3 s (Yarbus, 1967) and in general varies between 0.2 and 0.8 s. These values are however computed in simple tasks, for example recognition of easy objects. For example, in (Castoldi and Duţă, 2012) it was found that in crisis response and damage assessment the average fixation duration is about 1 second (998 ms) while trying to asses the building damage level in satellite and aerial images with high density features. In the case of SAR imagery it is not yet known the typical fixation duration employed by experts charting sea ice.

25 In our study we focus on the visual interpretation of SAR images with sea ice content, aiming to demonstrate the possibility of using eye tracking in this field, where monitoring and measuring the quality of the sea ice information by visual means of classification has also a great impact in safety of winter navigation.

## 2 Study Area, Datasets and Experiment Subjects

Our study area is the Baltic Sea of Fig. 2, located approximately between the 54° and 66° of northern latitude and between 30 10° and 31° of eastern longitude. Most parts of the Baltic Sea have seasonal ice cover and ice information is necessary for the winter ship navigation. This information is provided in the form of daily ice charts. The FIS ice analysts have experience of



sea ice charting from the SAR imagery since the availability of the ERS-1 SAR imagery in early 1990's. The analysts are also familiar with typical Baltic Sea ice conditions and are able to provide valuable insight regarding the ice charting process.

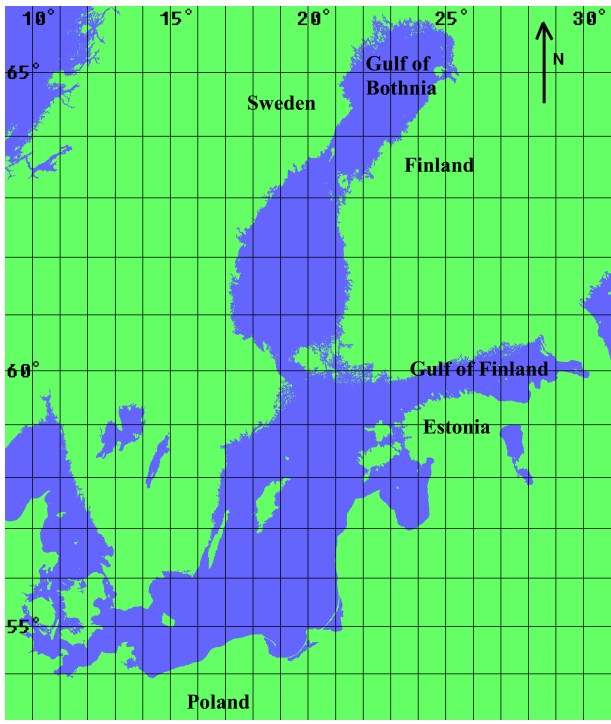

**Figure 2.** Baltic Sea, the study area.

## 2.1 Stimulus images

We used five RADARSAT-2 (RS-2) ScanSAR Wide images covering different regions of Baltic sea across three different winter seasons. The RS-2 SAR images with their acquisition time and central coordinates are listed in Table 3. The images were selected so that different ice conditions were present from easy to recognize open-water and sea ice areas, to complex texture patterns. They are also shown in 3.

In FIS, the original SAR images are typically reduced in size for easier manipulation and saving disk space. This reduces the amount of detail available for analysis. The 100 m resolution of the SAR imagery used by the FIS analysts is lower than the original resolution. Here we used the same down-scaled resolution for the RS-2 SAR images.

Due to the complexity of the visual SAR texture features of sea ice images, we included for reference five non-SAR images with easy-to-recognize content: a human face, a flower, a fish, a cat and a bird. These are referred to as natural images. They were selected from a publicly available eye tracking database (Judd et.al, 2009). Natural images were used as stimuli to establish a reference gaze behaviour for the analysts when viewing images that they are familiar with, with easy to recognize objects, regardless of their expertise level.





## 2.2 Experiment Subjects

In total four persons took part in the study. Of these four, two were experts (denoted as E1 and E2) in the SAR sea ice analysis; E2 with an experience of ten years, and E1 with over 25 years in operational ice charting and other projects related to the sea ice analysis using SAR and other imagery, see Table 3. Experts are hard to recruit, as on a national scale there may be only a few available.

The experts have acquired their experience and developed their skills in ice charting and SAR sea ice analysis through various ways. E1 focused the analysis mainly on the Baltic Sea area, having worked at the Finnish Meteorological Institute (FMI) already when the first SAR data became available for ice charting in early 1990's, while E2 focused mainly on the Caspian Sea and Greenland waters, when working at Danish Meteorological Institute (DMI).

Two novices (N1 and N2) with little or no familiarity with classification of SAR sea ice imagery participated to the study. N1 had been familiar with Geographical Information System (GIS) software and was a contributor to the development of FMI's own sea ice charting software. At the time of recording N1 was working under FIS and had very little practice on the ice charting, but was somewhat familiar with the ice charting practices and SAR analysis. N2 had no experience with the SAR sea ice analysis, but was somewhat familiar with remote sensing of sea ice in general.

## 3 Methods

### 3.1 Hardware and software for Eye Tracking Experiment

To record the eye movements, we used a Tobii X2-30 Eye Tracker (30 Hz) (Tobii Technology AB, 2014), in connection with an ordinary PC and an external monitor with a 22" diagonal size, similar to the ones used in the operational ice charting at the time of recording. Verbal protocols were recorded for the entire session, and used later for data filtering and analysis. All users gave their consent for recording their voice and eye movements.

We instructed the participants to look at the selected images and interpret the content verbally. For the non-SAR images, the participants were instructed to recognize the objects from the background by naming the category which belongs to and give a short description of the object. When looking at the SAR images, the participants had the task to describe their content freely by identifying sea ice types and features and classify them as they would in a typical ice charting routine.

To display the stimuli and record the eye movements data we used the Tobii Studio software (Tobii AB, 2016). Due to the differences in complexity and image size and resolution, natural and SAR sea ice images were viewed using different tools. The natural images were shown through the Tobii Studio recording software as image stimulus, while the SAR images were opened and viewed with an image viewing program (Irfan View) which allowed users to freely change the scale or pan the viewed images.





## 3.2 Eye Tracking Experiment Procedure

At the beginning of each recording session (one session per user), the users were required to perform a 9-point calibration routine, by focusing their gaze onto the displayed calibration points in predefined position on the screen. After a good calibration was obtained, the recording started with a page showing lines of text with instructions for the user. Then, the images were
displayed one by one in an alternate manner, so that a natural image was displayed before each SAR image. There was no time limit for displaying an image and each user had the power to take their time visually inspecting each scene. The users were asked to manually change the displayed stimulus when finished analysing it, by simply pressing a keyboard key or using the mouse by clicking the Close button.

## 3.3 Data analysis methodology

Eye movement data, verbal explanations and the interaction with the shown imagery were used in our analysis.

Here we study how the two experts perform in identifying and classifying sea ice regions in the SAR imagery without any additional information. More specifically we want to see if there are clear differences between their results and how could these be related to their experience and training.

We investigate following research questions:

1. Are experts able to identify/classify ice types in the SAR images without additional ice information? If yes, which ice types/features they can identify based on SAR data alone? Are they in agreement with each other and with the official FIS ice chart? 2. What visual strategies, if any, they follow during the visual interpretation of the sea ice features in the SAR imagery (e.g., order preference, time to get familiar with each SAR image before the actual analysis)? 3. Can eye movements distinguish between experts and novices? 4. Are the eye movements of experts during SAR analysis any different than when they look at
the natural images, or can their gaze distinguish between easy or difficult to interpret areas?

We divided the gaze data into segments that correspond to the scanning phase and the analysis phase. The scanning phase refers to the beginning of each image visualization, when users first look at the SAR images just before starting their analysis. The analysis phase refers to the time when the users have started their analysis by providing verbal explanations and focusing their attention on one region at a time. We computed the average dwell time, fixation duration mean, standard deviation, and
fixation density (number of fixations per ice area).

**Table 2.** Participants in the eye tracking experiment.

|  | E1 | E2 | N1 | N2 |
|---|---|---|---|---|
| age | 56 | 39 | 28 | 24 |
| experience | 25 | 10 | 1 | 0 |
| visual defects | no | no | no | no |

*E* = sea ice analysis expert; N = novice; experience = nr. of years





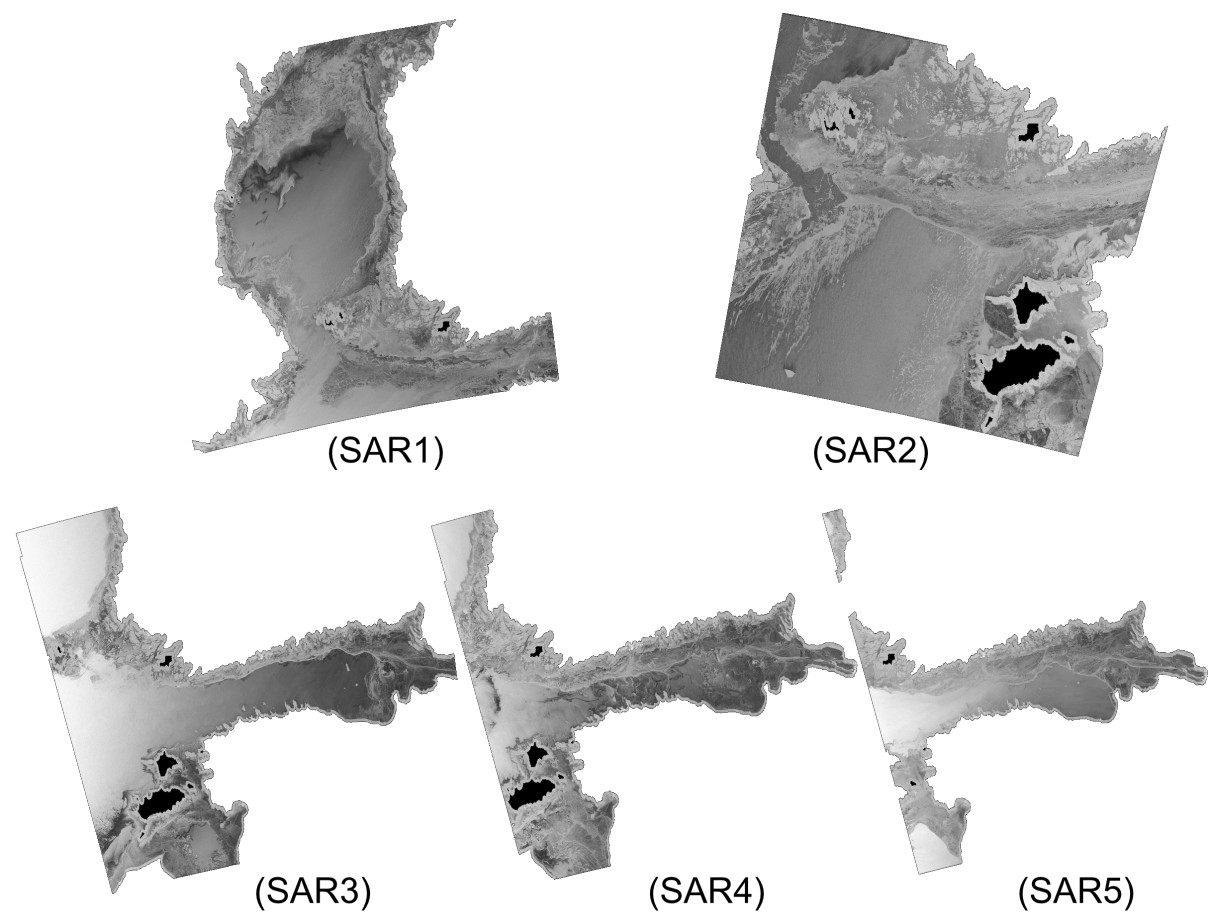

**Figure 3.** RADARSAT-2 SAR images listed in the Table 3

**Table 3.** RADARSAT-2 ScanSAR images over the Baltic Sea used in the eye tracking experiment.

| Image | Date | Time (UTC) | Latitude | Longitude |
|---|---|---|---|---|
| 1 | 2010-02-12 | 16:16 | 61.252003 | 19.829067 |
| 2 | 2011-02-27 | 05:04 | 59.394806 | 21.461995 |
| 3 | 2012-02-06 | 15:59 | 60.009427 | 24.561313 |
| 4 | 2012-02-13 | 15:54 | 59.986025 | 25.619694 |
| 5 | 2012-02-20 | 15:50 | 60.219841 | 26.569200 |

central image coordinates are given in WGS84.



## 4 Results

First, we discuss the qualitative observations and interaction with SAR imagery by the four analysts. This includes the SAR image analysis strategies employed by the two experts. Moreover, the gaze behaviours and performance measures are reported, more specifically we discuss the actual classification results by the two experts, including similarities and differences in the sea

ice recognition / classification results. Further, we look into how the two experts performed when compared to the official ice chart. Lastly, we look into the fixation statistics in relation to the SAR image complexity.

### 4.1 Qualitative observations and interaction with SAR imagery

The first difference between experts (E) and novices (N) was noticed right away, based on their reactions when they were shown the first SAR image.

When a new SAR scene was shown, experts were first visually scanning the entire image fore few seconds followed by the identification of the geographical region present in the SAR image: harbours, islands or other useful location information or features. Novices spent more time before they proceeded with the actual analysis. Even if novices recognized fast (in the first five seconds) that the image shown is a SAR image and eventually also the geographical area covered by it, they spent longer time before they started describing sea ice features. They also talked in a slower rate indicating confusion or lack of knowledge

(difficulty in recognition of the sea ice features or lack of confidence).

Novices' gaze was random at first, jumping fast from one sea ice area to another, including open water areas or areas with very thin/fresh ice. They seemed confused about what they should look for in the image. They also use more general terms for ice rather than specific ones (e.g., "some ice" or "thick ice" instead of *fast ice*) indicating non-familiarity with the region or with the ice charting terminology. Novices also made excessive use of zoom and pan tools available, but this did not help

much, their answers being confusing and limited. In general, the two novice users focused their gaze especially onto the most obvious (salient) features in the images, such as islands, ship tracks or clear ice boundaries.

Because the geographical span and the ice conditions varied with each SAR image shown, the experts also changed the viewed scale of the stimuli by zooming-in and out. For the SAR images 1 and 2, E1 did not change the scale at all, the ice covered areas being visible and distinguishable. For the SAR images 3,4 and 5, E1 did change the scale from 16 % to 28 %,

21 % and 21 % respectively. In these cases, the ice covered areas were much smaller compared with the SAR images 1 and 2. For all five images, E1 increased the scale 0.9 times the initial scale, in average per image. On the other hand, E2 required much more increased scale for all of the images, although he was able to provide some general description of the ice situation at the default scale. He did increase the scale in each image several times, reaching an average per image of five times the initial scale.

With the increase of the scale, only part of the image remained visible, thus forcing the users to change the viewed area by panning the image (almost constantly at higher scales). In general, a higher scale (i.e., the smaller area viewed) corresponded with a higher number of change of view (COV) times. To see how different participants reacted in average per image, we





computed the average COV rate (number of COV times per image), average zoom rate (AZ) and average view (AV) duration, see Table 4.

**Table 4.** Viewing measures computed for all five SAR images

| User | ACOV rate (%) | AZ rate (%) | AV duration (s) |
|------|---------------|-------------|-----------------|
| E1 | 5 | 2 | 108 |
| E2 | 22 | 6 | 349 |
| N1 | 49 | 15 | 111 |
| N2 | 9 | 9 | 177 |

ACOV = Average change of view; AZ = Average zoom; AV = Average view.

In average, the zoom rate for all SAR images is smaller for the experts with 2% and 6% while the novices have an AZ rate of 9 % and 15 %. Clearly the experts have a lower zoom rate which can be explained by the familiarity with the type of scenes
viewed, experience and speed of execution. By looking at the ACOV rate, we notice that E1 and N2 stay below 10 %, followed by the E2 at 22 %. N1 reached the highest score of 49%. This shows that N1 spends very little time for the actual analysis. The AV duration is the shortest for E1 and N1 with only 108 and 111 s, respectively followed by N2 with an AV of 177 s per image. E2 has the longest view duration of 349 s per image. The AV times include the identification of geographical elements in the images. Both experts keep a low AZ rate value, but E2 has a higher ACOV rate percentage of about four times larger than E1
and also the longest AV duration.

We found a difference in the way the two experts used the zoom. In average, E1 viewed the SAR scenes at a scale of 23 % in average, with a minimum scale of 15 % and a maximum of 28 %. E2 viewed the SAR scenes at a scale of 57 % in average, with a minimum of 15% and a maximum of 78 %. Here, we consider that a scale of 28% or lower corresponds to the default scale (DV) while higher values correspond to increased scales (ZV). Therefore, E1 viewed all the ice regions at the default scale,
with 23 % in average. E2 had a default viewing scale at 18 % in average and an increased viewing scale at 78 % in average. To understand better the significance of viewed scale values, it is worth to mention that images viewed at the default scale would be seen completely, while an increased scale meant that some of the content could not be visible on the screen at all times.

### 4.1.1   SAR image analysis strategies

For E1, the identification of ice types in the SAR images was carried out in general from the most dominant or challenging
ice area and slowly moving towards more easy to navigate areas. In most images he started the analysis with *fast ice* found close to land, followed by *very close ice* or *close ice*, ice boundary which may include a brash ice barrier (an obstacle for ship nagivation). In all cases, most of the *ice channels* were identified with their connecting harbour locations. *Open water* and *new ice* were also identified during the visual inspection but E1 spent very little time over these easy to navigate areas.

In a different manner compared to E1, E2 started to identify first the *open water* and *new ice* areas and then moved to
thicker ice areas (i.e. *level ice*, *fast ice*, *close ice* and *very close ice*). Similarly to E1, E2 also identified *ice channels* with



their connecting harbours when present, brash ice barriers and other ice features. A high level of detail was preserved by E2, spending more time describing smaller scale features than E1.

Both experts provided estimates of ice concentration and ice thickness, but not in a consistent manner to be able to include those into our analysis.Interestingly, the analysts did not identify the same number of sea ice areas or features. E1 identified a total of 49 ice regions, from which 31 correspond to an ice charting category, 17 correspond to ice channels and three to brash ice barriers. E2 identified a total of 109 regions, some of which were looked at multiple times and at different scales. From these, 88 regions correspond to an ice charting category, 11 to ice channels, and also three to brash ice barriers.

## 4.2 Gaze behaviours and performance measures

### 4.2.1 Natural images

By dividing the image content in object and background areas, we were able to compare the gaze data of all users while looking at the natural images, see Table 5.

**Table 5.** Fixation duration mean, standard deviation, fixation duration sum and the number of fixations are the computed gaze measures for the two categories of users while viewing natural images. The measures are computed over the foreground objects and background areas.

| Expertise | Focus area | N | FDM (s) | STD (s) | TFD (s) |
|-----------|------------|-----|---------|---------|---------|
| E | background | 203 | 0.43 | 0.31 | 86.63 |
| | object | 442 | 0.40 | 0.27 | 178.13 |
| N | background | 75 | 0.33 | 0.14 | 25.06 |
| | object | 296 | 0.43 | 0.35 | 128.63 |

FDM = Fixation Duration Mean; N = Number of fixations; STD = standard deviation; TFD = Total Fixation Duration.

The results show that there are no major differences in the way participants inspect an image with an easy to recognize object. Their task was to identify the objects in the scenes and label them. All participants had similar values for the fixation duration mean with an average of 0.4 s for both object and background areas. The only significant difference between experts and novices was in the number of fixations, which was much higher for the experts. This can be due to the longer time of analysis for the experts, spending on average 17 s of fixation per background image and 36 seconds per object image. Novices spent on average only 5 s per background image and 26 s per object image.

The higher number of fixations for the experts was mainly due to the longer time of analysis. This was in average 3.4 times more than the novices' for the background areas and 1.4 times more over the objects. Longer viewing time is also carried with a greater amount of details given by users during analysis.

During the analysis of the natural images, individual differences were observed amongst all four participants, but in overall the findings are consistent with existing work (Yarbus, 1967).



### 4.2.2 Experts and SAR image sea ice classification

When the SAR images were first shown to the analysts, they first looked at the image content before starting the verbal explanations (analysis). Consequently, we have divided the gaze data into scanning phase (before they start the analysis), and the analysis phase.

During scanning phase, both experts required on average 0.4 s of fixation duration mean, same time as for the natural images. This shows that while they're getting familiar with the viewed scene, they don't require additional cognitive effort. However, they required different amount of time for the image scan. E1 required on the average 5 s of fixation time per image while E2 required on the average 7 s. During the scanning phase, the analysts identified the type of image shown, geographical area, and even some of the ice features. Their gaze paths during the scanning phase show individual differences. For example, E1
seemed to have a denser gaze onto the regions with higher risk for navigation, while E2 had its gaze covering a wider range of sea ice types and geographical regions. Figure 4 shows an example of gaze data (fixations) collected for both experts during the scan and analysis phases of the SAR image 2.

Interestingly, experts' gaze during image scan shows a preview of their gaze during analysis phase, except they have only fewer gaze points in each area. They both seem to have looked at and analyzed the same areas they previewed during the
scanning phase, including the problematic areas (i.e., sea ice regions which were more difficult to classify and which were not classified in agreement between each other). This means, that in the first few seconds of looking at a SAR image, experts already identify the main sea ice regions to classify.

**Table 6.** Fixation measures computed for Expert users while inspecting the SAR images during first scan and analysis phase

| User | Gaze Activity | $\overline{FD}$ (s) | $\overline{FDM}$ (ms) | $\overline{STD}$ (ms) | $\overline{N}$ | $TFD$ (m) | $N$ | $NF$ |
|------|---------------|------|-------|-------|----|-----|-----|-----|
| E1 | scan | 5 | 391 | 193 | 14 | 0.45 | 70 | - |
|    | analysis | 5 | 674 | 521 | 10 | 4.45 | 476 | 49 |
| E2 | scan | 7 | 404 | 157 | 18 | 0.56 | 90 | - |
|    | analysis | 8 | 630 | 479 | 13 | 14 | 1443 | 109 |

$a$verage values($\overline{n}$) are computed per area/feature; NF = Nr. of areas/features identified.

During the analysis phase, experts focused at one ice area at a time, spending 1-3 fixations when switching between regions. In some cases, they were re-scanning the viewed scene or part of it in between two consecutive analysis segments. In some
cases, in addition to the sea ice categories identified in the SAR images, both experts provided detailed information such as estimates of sea ice thickness or concentration, as well as hypothesis on the ice formation or melt. Compared to E2, E1 spent less time fixating on each SAR image while providing detailed information. On average, total fixation duration for E1 was about 62 s per SAR image and only 5 s per ice area identified, while E2 had in average 5 minutes of total fixation duration per image and 14 s per ice area identified.

This is an interesting result, showing that the experts need only few seconds (5-14) of fixation time to be able to classify an ice area. Also it underlines a difference in style of analysis. E2 spent in average about five times longer analyzing an image and

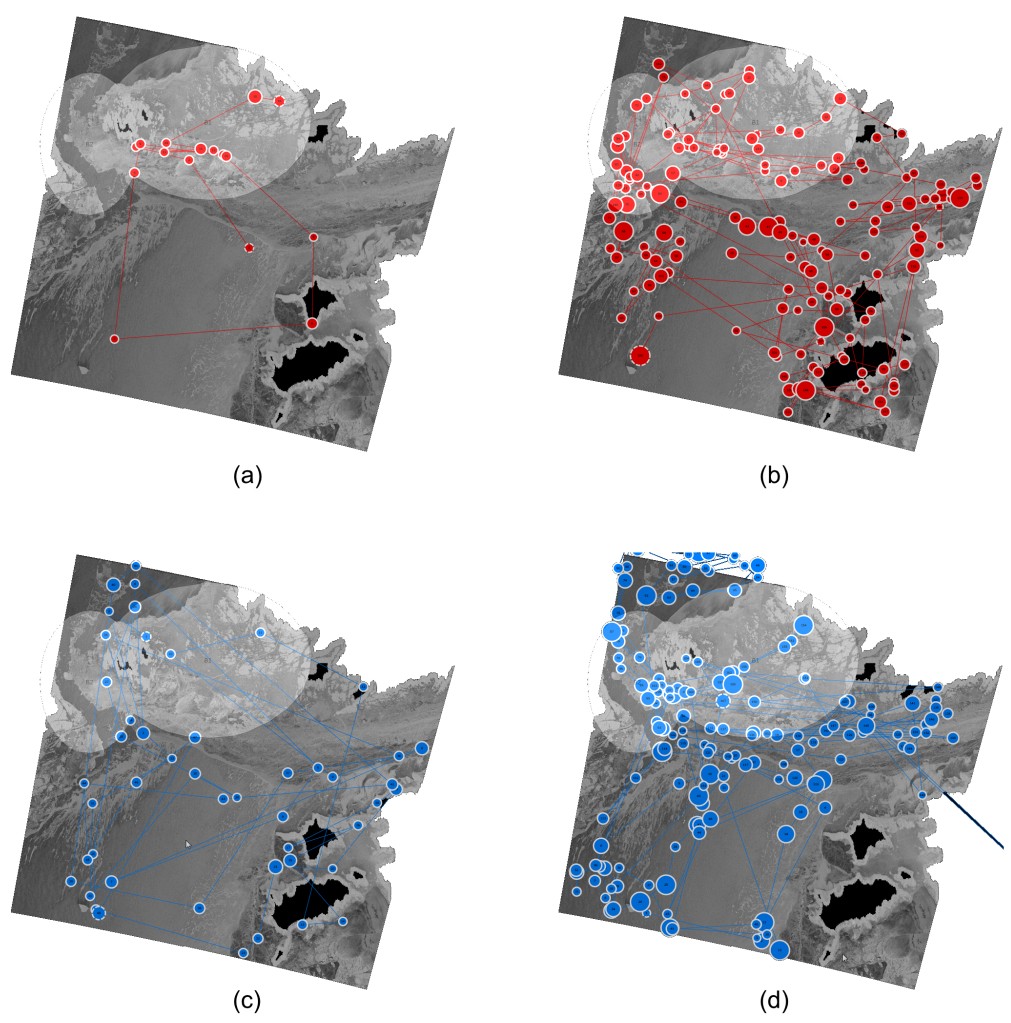

**Figure 4.** Gaze plots of the two experts, during visual scanning phase (left) and analysis phase (right) of the RadarSat-2 SAR image 2 acquired over the Åland Sea and the Northern Baltic Proper. Regions with higher priority in analysis are also highlighted. Here the gaze points are represented by circles of size corresponding to the fixation duration and have one color for each analyst (red for E1, blue for E2). Here, the time they spend scanning the image differs considerably (the most, compared with other SAR images). E1 spends only 5 s while E2 spends 18 s. This is, however, an exceptional case for E2, whom expressed clearly that this particular image was more difficult to interpret.





about three times longer analyzing an ice area than E1. These differences could be explained by the inequality between the two experts in terms of years of experience and of training. The more experienced analyst (E1) shows faster scanning and analysis times (see Table 6).

Looking further into the actual recognition of the ice types and features, we divided the data into segments corresponding to

5 each ice type or feature identified by the two experts, following the definitions typically used in the Baltic sea ice charts (see Table 1).

In addition to the ice categories, both experts identified other important features such as ice edge, brash ice barriers, ice channels, leads or ridges. We included *ice channels* and *brash ice barriers* along with the ice categories in our results, because these are clear and recognizable features and extremely relevant for the ice navigation. Also, these two ice features were

10 identified more than once by both experts during analysis.

The ice types and features identified for each segment and their computed eye movement measures are presented in Table 7. The gaze measures reported here are given as average value per an ice category.

**Table 7.** Fixation measures per ice category/feature computed for the two experts while inspecting the SAR images

| Participant | identified ice category | $\overline{dwelltime}$ (s) | $\overline{FDM}$ (ms) | $\overline{STD}$ (ms) | $\overline{N}$ | $NSAR$ | $NF$ |
|---|---|---|---|---|---|---|---|
| E1 | *open-water* | 4 | 721 | 654 | 8 | 3 | 7 |
| | *very open ice* | 6 | 473 | 250 | 13 | 1 | 1 |
| | *open ice* | 5 | 385 | 164 | 13 | 1 | 1 |
| | *close ice* | 5 | 471 | 368 | 10 | 1 | 1 |
| | *very close ice* | 9 | 638 | 517 | 16 | 3 | 5 |
| | *new ice* | 5 | 454 | 236 | 10 | 4 | 4 |
| | *level-ice* | 8 | 789 | 614 | 13 | 3 | 6 |
| | *fast ice* | 8 | 520 | 276 | 17 | 2 | 3 |
| | brash ice barrier | 6 | 482 | 240 | 12 | 3 | 3 |
| | *ice channels* | 4 | 788 | 658 | 5 | 5 | 17 |
| E2 | *open water* | 9 | 733 | 590 | 15 | 5 | 15 |
| | *very open ice* | 7 | 485 | 170 | 17 | 1 | 3 |
| | *open ice* | 6 | 498 | 260 | 11 | 2 | 5 |
| | *close ice* | 16 | 956 | 755 | 32 | 2 | 3 |
| | *very close ice* | 7 | 549 | 347 | 15 | 5 | 19 |
| | *new ice* | 6 | 674 | 533 | 11 | 4 | 16 |
| | *level ice* | 5 | 601 | 472 | 10 | 5 | 14 |
| | *fast ice* | 11 | 562 | 389 | 16 | 5 | 13 |
| | *brash ice barrier* | 10 | 578 | 349 | 18 | 2 | 3 |
| | *ice channels* | 6 | 764 | 641 | 9 | 4 | 11 |

*av*erage values are computed per viewed ice area; N - fixation density (number of fixations recorded in an ice area); NSAR = Nr. of SAR scenes; NF = Number of features / areas.



From all of the ice types identified, we found in average a FDM difference of 76 ms between the two experts, with an average FDM of 556 ms for E1 and 632 ms for E2. However, for E2 we included here the FDM values during both default view and increased scale view. If we take into account only values at the default scale for E2, the average FDM per ice category identified increases to 720 ms and the difference between the experts increasing from 70 to 164 ms. This means that for E2 it was more

5    difficult to recognize / classify some features at the default scale, and by zooming-in, the recognition effort decreases. At the default scale, most difficult to recognize / classify ice categories for E2 were new ice, open water, very open ice, and very close ice, from highest to lowest, respectively. Overall, the largest difference between the FDM data of two experts analyzing different ice categories were found over close ice, new ice and level ice regions, see Figure 5.

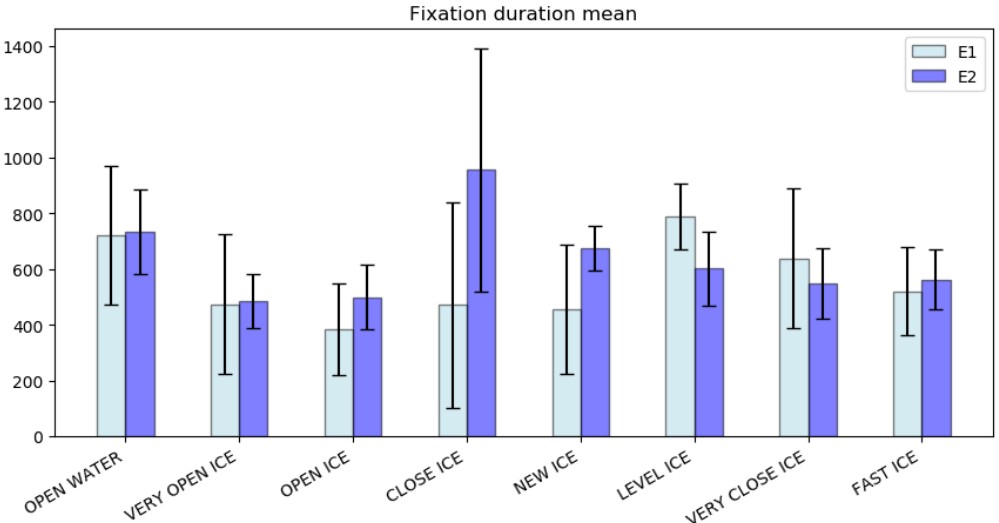

**Figure 5.** Average fixation duration mean with standard error (SEM) values recorded over recognized ice categories for experts. The ice categories are arranged based on the increasing ice concentration values from left to right.

These values are higher than the values recorded while viewing natural images, indicating a rise in the cognitive effort

10    required for the identification and classification of more complex SAR features. The FDM values recorded for both analysts vary between a minimum of 385 ms and a maximum of 956 ms, indicating that the level of complexity in perception for each ice category is different. In general, higher FDM values were recorded during analysis of ice regions richer in features, and thus, potentially more challenging for navigation. In contrast, lower FDM values were recorded mostly over more uniform and easily navigable areas. For example, *fast ice* which grows close to the land and has homogeneous backscatter spatially is easily

15    recognizable visually. Here, an average FDM value of 520 ms was recorded for E1 and 562 ms for E2.

The difficulty in interpretation of sea ice features in the SAR imagery is dependent on the level of expertise, training practices and personal style of analysis. This can also be seen from the slight variation of the FDM values recorded between the two experts over different ice categories or features. For example, E1 recorded the highest FDM value of 789 ms over *level ice*



category, while E2 recorded the highest FDM value of 956 ms over *close ice* category. The high FDM values recorded for E1 over *level ice* areas may be due to the fact that the analyst had more difficulty in estimating the overall thickness of the ice cover in those areas. For example, some of the *level ice* areas may be present within leads which seen at small scale may require longer time for interpretation. The lower FDM value of 601 ms recorded over *level ice* for E2, may be explained by the fact

that this analyst had analyzed the SAR content in both default and increased scales. If true, than it can be assumed that not all features can be easily recognized at lower scales, but seen larger they may require less cognitive effort for the classification. A direct consequence of lowering the effort of recognition may relate to a decrease in uncertainty in classification of a viewed ice region. On the other hand, the high FDM values recorded over *close ice* areas by E2 may be due to the difficulty in estimating the ice concentration, especially when the areas consist of a mixture of unknown ice thicknesses. For both experts, the lowest

FDM values (constantly less than 500 ms) were recorded over *open-ice* and *very open-ice*.

Interestingly, when identifying sea ice channels, both experts recorded an average FDM of 776 ms. This suggests that both had the same level of difficulty in identifying the ice channels feature. This can be due to the extremely small scale at which these features were viewed. *Ice channels* typically consist of very narrow lines indicating the ice-breakers routes that extend from one port to another. The presence of vessels along these routes may help analysts in the identification of such features.

Here, longer fixations over the ice channels could be explained by the small size (i.e., one to several pixels) of the vessels which need to be recognized. Yet, analysts need only few fixations (seven in average) to identify the ice channels.

Somewhat in contrast to the ice channels, brash ice barriers are easier to identify because they lie at the boundary between ice cover and open water or ice free region. This is confirmed by the experts having recorded lower FDM values during brash ice barriers identification, such as 482 ms for E1 and 578 ms for E2. Here, they recorded a higher fixation density of 15 fixations

in average. Both experts identified the same number of brash ice barriers (three) but they had different number of ice channels identified (17 for E1 and 11 for E2).

Contrary to our expectations, both experts had relatively high FDM values (721 ms for E1 and 733 ms for E2) over *open water* regions. This can be explained by the occasional presence of broken ice floes or rough sea ice conditions resulting in a mixture of very small and very high backscatter.

Both experts had the longest dwell time (i.e., the visual time spent classifying an ice category or feature) while looking at the most feature rich areas, see Table 7. The dwell time per ice category or feature viewed varied between the two experts. E1 spent in average 9 seconds over *very close ice* and 8 seconds for both *level ice* and *fast ice*. E2 spent the longest dwell time of 16 seconds interpreting *close ice* category. E2 also recorded 11 seconds over *fast ice* region. Even though, *fast ice* may present in general a more uniform surface, in some cases the presence of islands, vessels and ice channels may keep the analysts' gaze

for longer in the area. *Very close ice* and *fast ice* regions were the two most dominant ice classes in all of the SAR images viewed.

In contrast, the shortest dwell times were recorded over areas that may be considered easy for navigation. E1 recorded the shortest dwell time of 4 s over *open water*, 5 s over *open ice*, *new ice* and *close ice* areas, and 6 s for *very open ice* areas. E2 recorded shortest dwell time of 5 s over *level ice* areas, followed by *open ice* and *new ice* areas for which he spent 6 s

in average. However, E2 required as much as 7 s of dwell time over *very open ice* and *very close ice* regions. This may be





explained by the fact that both ice categories may consist of higher amount of features, either ridges, leads or cracks in the *very close ice* pack or a mixture of ice floes of different thicknesses in the *very open ice* regions.

Interestingly, E1 identified only one *close ice* region which had similar gaze values as for the easy to navigate areas while spending also very little time analyzing it (5 s). In contrast, E2 identified three regions of *close ice* for which he also recorded

the longest FDM and the highest number of fixations compared with all other ice categories identified. These differences may have occurred due to the extra time spent analyzing the SAR images together with the increase in scale used by E2. Nonetheless, this result underlines the subjectivity and difference in perception and visual interpretation of the SAR image sea ice features when no additional information is available to the analysts.

To see whether the increase of scale used by E2 affected his gaze and subsequently its difficulty in identifying and classifying

ice types in the SAR imagery, we compared the average FDM values recorded over the recognized ice categories during default view (DV) and zoomed-in view (ZV) for all five SAR images.

The average fixation duration mean computed during default scale is higher (7xx ms) than the one during increased scale view (5xx ms). Out of 88 sea ice types identified by E2, 37 ice regions were identified during default view with an average FDM of 720 ms recorded per ice type. 51 ice regions were identified during zoomed view with an average FDM of 551 ms

per ice region. This slight increase in FDM values recorded for the default scale view may relate to the increase in difficulty in ice type recognition when the SAR image is viewed at lower scale. Again, to show that dwell time is not necessarily related to the difficulty in the ice type recognition, here the computed average dwell time was 8 seconds per ice type for both default and increased scale view.

### 4.2.3 Discrepancies in ice information described by the two experts

Beside the level of detail and duration of scanning and analysis of the presented SAR images, there are also discrepancies in the ice information depicted and described by the two experts while looking at the same sea ice features in the SAR images. One of these examples is discussed here.

During the analysis of the SAR image 5, E2 focused on an area (A-d), which seemed more difficult to analyze than other areas. The difficulty was shown not only by longest time spent over it (33 s) but also by the verbal explanation associated with

it. This segment of analysis, is deconstructed in Figure 6 where the gaze plots associated with each identified (or unidentified) ice region are shown.

First sub-segment (1) constitutes of 18 fixations recorded in 8 s. From these, 12 points are concentrated around an area with uniform backscatter and no distinct ice features, having a similar texture and contrast as the open-water area present in the southern Gulf of Finland. Here, the analyst notices an opening in the ice pack, but is unable to classify it as ice or open water

due to lack of high contrast features. Then, he moves his attention to two more similar openings in vicinity (2), identified this time as *open ice* in 7 s. At this moment, the analyst decided that these openings could belong to a larger area (3) that includes also the first identified opening, by virtually drawing its contour with the mouse cursor. Then, another polygon (4) is identified and shown with only five fixations in 4 s, polygon which was not classified nor well defined its contour. Lastly, the polygon



**Figure 6.** RadarSat-2 SAR image 5 acquired over the Gulf of Finland. Upper left corner: Original image; a) Zoomed-in view (130%) as seen by E2; b) gaze plot of E2 over an undefined sea ice area, the gaze plot here totals 33 seconds; this plot is divided in five sub-segments as follows: 1 - the identification of an undefined area; 2 - identification of two smaller openings (*open ice* or *very open ice*); 3 - the contour of one ice class region; 4 - the contour of another ice region; 5 - re-evaluated contour 3, identified as *open ice* or *very open ice* (3-4/10). c) rough contours of each identified region, drawn by the authors.

is virtually re-drawn while becoming polygon 5, classified by the analyst as *open ice* or *very open ice* 3-4/10 during 10 s of analysis.





By looking at these sequences of data, a clear difference in the gaze density of E2 occurred for the case when he is not able to classify an area (1) and for the cases when he does (2-5). A higher gaze density occurred while trying to classify region (1), while a lower gaze density was recorded for the other areas. The most comparable are sequence (1) and (2), where the analyst is classifying similar type of openings (*open ice*). Here, he spends about the same amount of time (8 and 7 s) but his gaze points

have a wider spread over the region 2 which he identifies as *open ice*. Region (3) and (5) are mostly constituted of fixations during the analyst indicating the polygon's boundary, which he identifies it with *open ice* or *very open ice*. In these cases, the points are distributed more equal and having similar duration.

**Table 8.** Fixation measures computed for E2 during analysis of the region with longer fixations, of the RADARSAT-2 SAR image show in Figure 6.

| segment | dwell time (s) | FDM (ms) | STD (ms) | N |
|---------|---------------|----------|----------|-----|
| 1 | 8 | 442 | 325 | 18 |
| 2 | 7 | 396 | 201 | 17 |
| 3 | 4 | 329 | 102 | 13 |
| 4 | 4 | 271 | 100 | 13 |
| 5 | 10 | 360 | 119 | 28 |

On the other hand, by looking at the average fixation duration mean and standard deviation values over the same region (see table 8), we notice that the analyst recorded here the lowest values (in average 360 ms for FDM and 169 ms for STD)

compared with all other ice types identified in all SAR images viewed (in average 659 ms for FDM with 485 ms for STD and a minimum of 549 ms for FDM and 347 ms for STD). This once again shows that when an area in SAR presents a high level of uniformity and lacks features with high informational value, the analyst will not spend longer fixations over that area trying to extract more information. This kind of result supports also our previously stated assumption that longer fixations correspond to those ice features rich in information that carry critical value to the analysis. On the contrary, E1 did not mention the A-d

region at all, nor spent any time gazing at it. Only few of E1' gaze points fell onto this area, during the whole image inspection. More precisely, we recorded only one fixation point during the DV scan of this SAR image, with a duration of only 332 ms and seven fixations during the ZV analysis, while describing the "*consolidated ice* or *fast ice*" region in vicinity. This can be explained by the fact that lack of features did not attract E1' attention, therefore E1 spent extremely little time on it without even describing it.

Long dwell time (33 s) over difficult area (Figure 6) shows that E2 was trying to find some relevant features that could help to describe better the viewed ice covered area but short fixations together with verbal explanation reveal the lack of value-added features, leading to a final classification of the area which does not completely agree with the classification made by E1 nor the classification found in the official FIS ice chart.



**Figure 7.** (a) An extract from the FIS ice chart (© Finnish Meteorological Institute) constructed with the RADARSAT-2 SAR image 5 and other data by two on duty analysts, other than the ones who participated in this study. Main sea ice boundaries drawn by E1 (b) and E2 (c) independently of the eye movement data collection, using only the SAR image 5.





### 4.2.4 FIS ice chart vs. Experts

To evaluate how the blind analysis of the SAR image sea ice features performed by the two experts would compare with the classification provided in the FIS ice chart, we asked the experts to draw the contours of the main ice regions and assign an ice class for each region in the SAR image 5. Their results are presented in Figure 7, together with the corresponding (next day) FIS ice chart (constructed by two analysts during their official duty with the SAR images used here as one of their several information sources).

The ice chart is showing the ice situation in the Gulf of Finland (GoF) contoured based on all sources of information available at that time. The region of frozen sea that separates the very thin ice and *open water* area from the *fast ice* in the northern part of GoF, is classified here as *very close ice*, rafted including two *very open ice* areas (same *open ice* areas were described by E2) and also few small areas marked as *level ice* (purple color). There is also two *open ice* areas marked with yellow color.

When comparing the ice chart to the ice regions marked by the two experts, we notice a general agreement between the two, especially for the most dominant ice types, such as land *fast ice* and *very close ice* covering significant area along the coast. Agreement is good over *open water* or ice free region in the south of GoF. However, disagreements are seen especially for ice types such as *close ice*, *open ice*, *very open ice* and *level ice*, where the analysts disagree. Not knowing the real sea ice conditions or the previous day situation, naturally explains these differences between experts' analysis and the classification in the FIS ice chart, especially for the lower ice concentration regions (open ice or less). However, in the case of open ice areas marked in the FIS ice chart, experts had contradictory interpretations, especially for the region close to the Estonian coast (low left side of Figure 6). E1 having marked the area as open water while E2 has marked the same area with very close ice. This is a considerable difference from the navigation point of view, since one expert is basically permitting independent navigation without the ice-breaker assistance while the other expert is restricting navigation with ice-breakers only.

### 4.3 Fixations statistics and their relation to SAR image complexity

We studied the dependence of the fixation duration for the two experts as a function of SAR image complexity by computing the local edge and corner densities in the SAR images. Edge and corner points in the SAR imagery were detected by applying an edge and corner algorithm based on local binary patterns described in Karvonen (2013). Because the locations of the fixations are often not exactly at the target we studied the numbers of edge and corner points within two different circular neighborhoods with radius of 20 or 50 SAR pixels from a fixation point. The fixation locations may not exactly match the centre of the region of interest because of possible measuring inaccuracies and also due to the fact that the human visual acuity at the foveal center corresponds to more than just one pixel at the centre of the region of interest, i.e., the foveal vision corresponds to approximately 2 degrees of the entire visual field (Duchowski, 2017). These comparisons were made using only the default scale images.

The results did not indicate very strong correlation between the fixation duration and the SAR image complexity. For the shorter fixation durations there was a large deviation in the image complexity. However, when divided into two duration categories of less than 500 ms and over 500 ms, the image complexity was about 10% higher for the longer duration category,





measured both by number of edge points and corner points and for the both 20 and 50 pixel radius. We also noticed that for fixations longer than 2000 ms the complexity was always above a certain limit but for the shorter fixations the range of the image complexity was varying a lot below and above the limit value.

5 We also analyzed the fixations statistics for each ice class looked at (the same sea ice categories presented previously in Table 7) by the two experts. We computed the average and standard deviation of fixations duration for each ice class, and also included the ratio of the total fixation time within each ice type segment and the ice type segment area. The segment area was defined by the convex polygon spanned by the segment points. The results of this analysis are shown in Table 9. It can be seen that in total E1 spent most time in recognizing *ice channels* and E2 in recognizing *new ice*. But when looking at the ratio of total fixation time to the segment area, the *very open ice* segments E1 seems to pay attention especially to *very open ice*

10 and E2 to *open water*. This is a surprising result as *open water* typically does not contain much features, i.e. the SAR image complexity for *open water* areas is typically low. However, distinguishing between thin or *level ice* and *open water* is often a difficult task and may require a lot of attention.

**Table 9.** Fixations statistics for the ice classes of the SAR image segments.

| Expert | N | average | STD | rel time | ice type |
|--------|------|---------|---------|----------|----------|
| E1 | NA | NA | NA | NA | *open-water* |
| | 13 | 472.54 | 250.26 | 0.52 | *very open ice* |
| | NA | NA | NA | NA | *open ice* |
| | 10 | 471.20 | 226.56 | 0.20 | *close ice* |
| | 37 | 483.86 | 257.70 | 0.22 | *very close ice* |
| | 12 | 484.08 | 231.25 | 0.21 | *new ice* |
| | 52 | 719.06 | 733.47 | 0.18 | *level-ice* |
| | 51 | 466.00 | 314.10 | 0.16 | *fast ice* |
| | 13 | 606.77 | 400.93 | 0.20 | brash ice barrier |
| | 19 | 894.89 | 1109.44 | 0.14 | ice channels |
| E2 | 151 | 663.48 | 619.91 | 0.31 | *open-water* |
| | NA | NA | NA | NA | *very open ice* |
| | 25 | 492.72 | 278.92 | 0.12 | *open ice* |
| | NA | NA | NA | NA | *close ice* |
| | 85 | 610.31 | 387.47 | 0.18 | *very close ice* |
| | 41 | 820.98 | 1060.46 | 0.25 | *new ice* |
| | 47 | 569.51 | 366.80 | 0.14 | *level-ice* |
| | 104 | 568.10 | 510.36 | 0.28 | *fast ice* |
| | 38 | 672.92 | 461.84 | 0.19 | brash ice barrier |
| | 47 | 694.38 | 649.13 | 0.19 | ice channels |



We also performed a SAR segmentation applying the Iterative Cumulative Model (ICM) algorithm (Besag , 1986) and then computed the number of edge and corner pixels within each segment normalized by the segment area (in pixels). These ratios characterize the local complexity of the SAR image. These were then visually compared to the total duration of the fixations within each segment, also normalized by the segment size. The resulting segmentation and the edge point, corner point images

and the fixation duration images for the experts E1 and E2 corresponding to SAR1 are presented in Figure 8. It can be seen that both E1 and E2 use relatively more time in the northern parts of the image with most of the sea ice. However, there also exist significant differences between E1's and E2's behaviors.The correlations between the quantities measuring image complexity (edge and corner densities) and the E1 and E2 segment-wise relative fixation duration were rather low, around 0.2, but still clearly positive.

One evident reason for the low correlation and correspondence of the fixations duration with the image complexity is that the fixation typically concentrate on the SAR segment boundaries, and thus often their location is within adjacent segments of the actual target segment.

## 5   Discussion and conclusions

In this study we demonstrated for the first time that the eye tracking methodology can be used to identify sea ice regions or

features in SAR images which are prone to human subjectivity, and therefore miss-classification.

While restricting our study to the Baltic Sea region, we asked two experts in the sea ice charting to look at a set of five RADARSAT-2 ScanSAR images and visually identify the sea ice types and features without any other sea ice information available to them. Experts were able to correctly classify the most dominant sea ice types in the viewed SAR images, such as *very close ice* and *fast ice* (i.e., large scale sea ice conditions). These are the two most restrictive ice categories for navigation,

allowing passage only with the ice-breaker assistance. On the other hand, differences in classification occurred especially for less restrictive ice classes such as *close ice, level ice, open ice, very open ice* and *open water*. While these areas are not restricted to independent navigation for lower ice class vessels, navigation in these ice conditions may be more challenging.

Eye movement data measures such as the fixation duration mean (FDM), dwell time and fixation density were found to be extremely informative and directly related to the difficulty in interpretation of sea ice types or features viewed. We found

that FDM values of experts analyzing SAR images vary from 0.4 seconds (similar value was recorded for easy to recognize objects in non-SAR stimuli) to as much as one second, depending on the features looked at. Higher FDM values were recorded especially over sea ice regions where experts had more difficulty in estimating the ice thickness.

Individual differences between the two experts can be seen as slight increase in the gaze values computed for E2 in contrast to E1. From all the gaze measures computed for the two experts we can conclude that E1 is faster and identifies the ice types and

features in SAR images with less effort than E2. This result may be related to the higher expertise level for E1 and differences in the ice charting training routines when compared to E2.

Our data suggested that experts rely on changes in uniformity / homogeneity in an ice covered region to better understand its characteristics during the analysis. Thus, the more complex features an ice region presents (i.e., the more non homogeneous

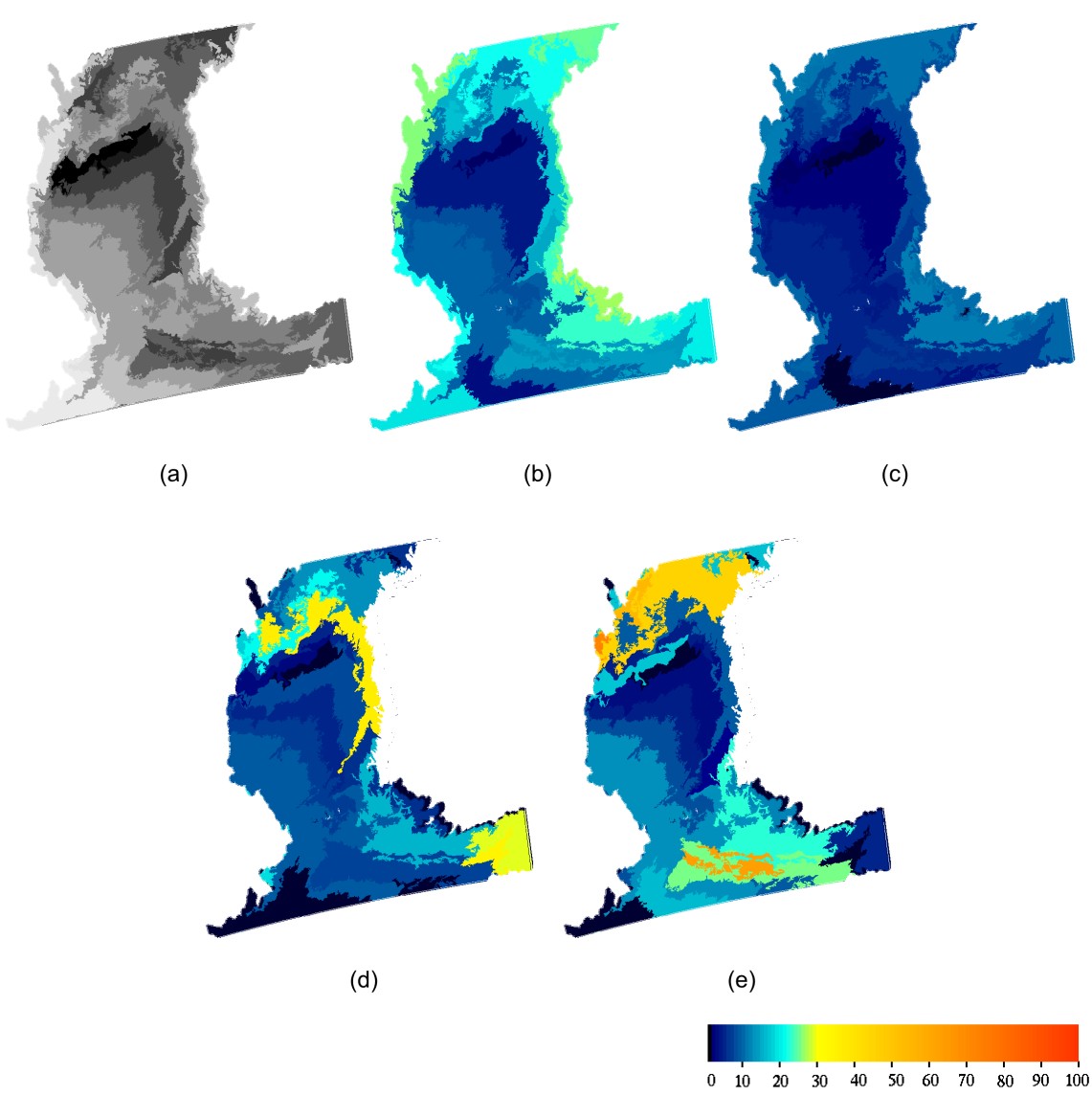

**Figure 8.** RADARSAT-2 SAR image 1 segmentation (a), edge density (b), corner density (c), ratio of the total fixation duration within SAR segments normalized by the segment size for E1 (d) and for E2 (e). The color of the images indicates the relative amount of edges and corners in percents for (b) and (c). The value for (d) and (e) is scaled such that it is $100 T_t/A$, where $T_t$ is the total time within a segment and $A$ is the size of the segment in pixels.

the region is), the more attention and effort is required by experts to classify it. Their cognitive effort which often mixed with visual effort of discriminating those features and associate them with characteristics of ice, can be identified by longer gaze





duration or larger fixation mean over a feature or set of features. The number of salient features in an ice covered area, their scale and geographical spread are also important factors influencing the eye movements of experts in the SAR image analysis. This study is the first to suggest a method of identifying the areas of sea ice in the SAR imagery with the lowest confidence for the analysts that are subject to miss-classification.

Interaction with the SAR imagery is also an important aspect in the ice charting that we looked at. In this case, we have asked also two novice analysts to join the experiment by visually interpreting the sea ice conditions in the same set of SAR images. In some cases users changed the scale of the viewed images several times during analysis. Novices rushed to increase the scale shortly after the images were displayed on the screen. At higher scales however, they still had difficulty in distinguishing between different ice types and features, but spending more time on panning and zooming the content.

Experts analyzing a SAR frame focus especially on key ice features that are meaningful for the navigation, such as ice edges, ridges, *lead*s and *ice channels*, while spending very little time or none over areas with uniform backscatter. This is in line with previous research on expertise showing that experts focus on relevant information, extending this knowledge to the SAR imageinterpretation domain. In contrast, novices had difficulties recognizing sea ice features and in many cases their answers were confusing, unclear and required more assistive questions. Their lack of expertise also reflected in an unfocused

gaze, spending significant amount of time looking at islands or ice floes confused as islands, ice edges or even homogeneous areas of *open water*. Their gaze also changed rapidly from one area to another, in many cases not concluding the analysis of an area with a clear ice definition.

Global warming and the thinning of the Arctic sea ice will result in sea ice conditions that will be more challenging to analyze. In future, ice charts will contain more of lower ice categories and less of large consolidated ice areas than currently.

Based on these facts, we conclude that manual ice charting requires more accurate classification of low ice concentration regions which may facilitate independent navigation, i.e. *open ice* and *very open ice*. The subjectivity and the source of miss-classification should be measured and recorded, so that future ice charts will be not only more consistent, but more reliable for safe navigation in sea ice covered oceans.

## 5.1  Limitations and Future work

Work presented here is a first effort trying to understand the eye movements of a sea ice analyst looking at a SAR image. The sample number in our study is low, and thus, the main findings are qualitative in nature. More quantitative and qualitative studies are required to better understand how sea ice parameters are visually interpreted by the experts and novices in the ice charting. Repeating this study with sea ice experts from several different organizations producing ice charts could reveal interesting insights into differences in the cultures of the SAR image analysis in these organizations. In general, we encourage

the use of eye movement data in further studies to deepen this kind of knowledge and to understand the uncertainties introduced by human analysts in the operational ice charting.

The aim of this study was to act as a proof of concept study. The deficiencies of the experiment design in this study can be improved based on the experience gained by this study for the related future research. More SAR data and more ice analysts are needed for a comprehensive study. And a more sophisticated software keeping account of all the zoom-ins, panning, cropping

and to record the fixation coordinates in the original full-scale image to enable a thorough analysis with all the fixation data integrated in the same master image.

Furthermore, eye tracking data acquired during SAR image analysis could indicate the complicated ice areas where higher resolution data is required or when the information available is not sufficient for a reliable classification. As a hypothesis,
long fixation duration are connected with larger uncertainties in the final ice charts. Because the ice charts at the moment lack uncertainty information, this is a very interesting topic that should be studied more.

The results presented here open new horizons for improving both manual and automatic analysis of SAR imagery for sea ice classification, but also for image classification in general, where the link between the observer and the automated method has not yet been established.

*Acknowledgements.* Authors would like to thank Tobii AB (former Tobii Technology AB) company for providing the eye tracker and the software for this study. Their software was also used in part for data analysis, and preparation of some of the examples shown. Authors would like to thank all the participants in the study for their contributions.

*Author contributions.*

The concept of the study was conceived by AG. JV was one of the expert analysts mentioned in this study who helped with
defining the research questions by providing insights into the ice charting practices and underlying existing challenges in the ice charting process. JV also selected the SAR imagery and helped with the interpretation of the results. The data collection was performed by AG, who also conducted the analysis. ER and RB also helped with the data analysis. AG, ER, JV, JK and MM contributed to the interpretation of the results and drawing the conclusions. All authors contributed to the writing and editing of the text.

*Competing interests.*

The authors declare that they have no conflict of interest.



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
