# Peer review of "Visual Interpretation of Synthetic Aperture Radar Sea Ice Imagery by Expert and Novice Analysts: An Eye Tracking Study."

_The Cryosphere, 2022_

## Author Comment (AC1)

**Author Responses to RC1: Comment on tc-2022-8", Anonymous Referee 1, 28 Mar 2022**

Citation: https://doi.org/10.5194/tc-2022-8-RC1

The Author Responses are marked with bold text.

**1   General comments by RC1**

After going through the first part of this paper I realized that there were already some fundamental flaws in the experiment. There was a lot of very good and detailed  information regarding the contrast between expert and non-expert analysts, as well as differences within these groups. However, the overall objective  of this paper and the relevance of these comparisons was not clear in the beginning, nor was it  throughout the manuscript.

**AR: This study was made mainly to find out could ET data be used to improve SAR based automated ice analysis and on what kind of details the ice analysts focus on in the imagery and how they are related to the SAR image statistics and resulting analyses. The novice analysts were included to find out the difference from how the experienced ice analysts work. We will consider simplifying the manuscript to make more clear the objectives of the study, taking into account the reviewer's comments and suggestions.**

Regarding the experiment design, in this study (and Karvonen et al 2015) the analysts are only shown a series of satellite images on Irfan view from different snapshots during the winter season. However, this did not include analysts having any of the ancillary  information that typically goes into an analysis, such as: knowledge of prior sea ice conditions (large-scale and regional), alternative ground-truth/observations (if used on these days), and prevailing weather conditions.

**AR: For this type of study it is actually better to reduce confounding factors; in vision studies, typically, stimuli images are used to understand effects of expertise. While this approach decreases the external/ecological validity, it significantly improves the internal validity of the findings - this is now the priority, as no previous studies have been done in this area. We cannot bring the whole software environment to the first study.**

**The experiment in Karvonen et al. 2015 was made in the international ice analyst workshop 2014 and the task assigned to the ice analyst groups was to assign SIC to predefined segments based on SAR imagery and then the results were compared to SIC estimated by an automated algorithm. SAR imagery are the main source of information for ice analysis and especially SIC is often estimated solely based on SAR imagery. It is not essential which software were used for image viewing in this the context or in Karvonen et al. 2015. The important thing is that the view is clear and can be used for ice analysis (or as in ice analysis). It is true that no ancillary information has been used in this study. Because the ice conditions are often changing, even rapidly in some areas, and the ice charts/analyses are made daily, or even**

**less often, ice analysts are used to make ice analysis based on SAR imagery alone (in addition knowing the acquisition** date **and time i.e. phase of the winter and typical ice conditions in the target area and the location based on the familiar coastline).**

The continuity that analysts have about environmental conditions is the core of the local knowledge that ice services rely for their routine chart production. The use of unusual hardware and software that puts the expert and non-expert analysts in unnatural situations to do an ice chart the way they would normally be done. The expert analysts are having to navigate a new setup that changes the way they habitually put together information for an ice chart analysis. Generally expert analysts have different systems between one another that allow them to be efficient to mentally collate all the information as they generate an analysis. For example, some will spend more time in the beginning reviewing information from the previous day(s) and others will focus on setting up the GIS layout. How this information is transferred between their own understanding and integrated in the analysis can vary greatly between analysts. Additionally, using a graphics viewer, Irfan Viewer, that is not the standard GIS that allows you to use familiar tools that expert analysts would be used to, that includes being able to access multiple sources of information and overlay them on one another, and have the same resolution they're accustomed, introduces a significant error in the design of these comparisons. If this were to be applied to analysts from other ice services we would expect to see a much greater spread. Fig. 7 shows these clear differences between the unnatural analysis using solely the image, and the actual ice chart which has much more information content.

**AR: We agree that the typical ice charting routines require historical data (e.g. previous sea ice chart) weather data, and other supporting data available as mentioned by RC1. However, in our study we focused on demonstrating a "proof of concept" in regard to the usefulness of the eye movement data of analysts in the field of ice charting. Therefore we focused strictly on the process of visual analysis of the SAR sea ice content disregarding in purpose the other sources of information which cannot be directly related to the eye movement behaviour. Using other sources of information would complicate the study too much at this point and we did not have the necessary resources. It is not essential which software is used to view the imagery if only SAR imagery are used. Irfan Viewer was selected because it was available and easy enough to use for the ice analysts. Even though GIS were not used in these experiments the ice analysts were aware of the locations of the SAR imagery based on the coastline shape and also the acquisition date and time. The idea was just to simulate the initial ice analysis used to sketch the first draft analysis based on SAR data and to track the analyst eye fixations during this process. This is assumed to be very similar as using the actual ice charting software.**

Regarding the issue of using this method to measure uncertainties of human perception, rather than trying to pick out a signal from noisy and subjective data, such as human bias, intercomparison studies between sea ice features in different products are a more accurate assessment of subjectivity and experiments are easier to carry out, especially when there's such a critical need to evaluate multiple skill sets of ice analysts from various international agencies. Studies like the Karvonen et al, 2015 and Cheng et al, 2020, do these types of comparisons on heavily processed datasets, where drawing the delineations between areas by the analyst is not required. Given that reference data is useful for quantifying the ability of automation to capture the variability

in sea ice, an independent variable in which to compare a real vs controlled situation is necessary when it comes to human subjectivity. There does not seem to be anything in the paper that supports the assumption that the areas where one tends to focus should correlate with lower confidence, thus higher uncertainty. Please refer to my previous comment above describing how analysts use information. The conclusion that analysts use more cognitive effort in areas where there is more uncertainty is also not convincing given the spread of the expertise included in the small sample size of this preliminary experiment. This same expert analyst from one ice service would not be expected to be as proficient in understanding ice conditions in different areas covered by other ice services due to different environmental conditions, such oceanographic and weather conditions, nor would they have expertise on the regional variability of the ice.

**AR: Thank You for the reference (Cheng et al. 2020).**

**Here the ice analysts performed their visual analysis based on SAR data lone, so in principle all the interpretation steps from the beginning are included, unlike in the references. It is difficult to say about the confidence because it is not measured objectively by any way. What we can say based on the collected data is in what kind of SAR areas the ice analysts eye fixations are located and where there are multiple fixations near each other. One target would be to utilize this information to develop better algorithms e.g. for SAR segmentation. Such algorithm could be based e,g, on detecting interesting targets relevant in ice analysis and segment boundaries.**

Despite the small sample size and relatively new approach that this proposed methodology could add in understanding uncertainties introduced by ice analysts, the initial outcomes from this case study does not add any value to the work already being done to resolve the issues of subjectivity in ice charts. Though the authors state in conclusions (5.1) this was proof of concept and they recognize a larger sample if needed, this current experiment design is not a reasonable method and complicates the evaluation process further because there are more variables that need to be taken into account regarding the expertise of the user and the amount of information that is available to them. This method would be especially challenging during the melt and summer seasons where the spread is going to significantly vary due to the geophysical limitations with the satellite sensors. Therefore, the continuity of the analyst needs to be taken into account, similar to weather forecasters, and the amount of time it takes for them to understand the situation should not be as significant a factor as how close the analysis is to actual environmental conditions.

**AR: We agree that more challenging sea ice conditions (e.g. melt season) the analysis becomes more difficult especially due to the limitations of the SAR sensors and availability of other supporting data. However, this is one important reason for conducting our study - the effect on the sea ice classification result when there is lack of sufficient supporting information. We basically wanted to see - what happens when analysts don't have other information regarding the sea ice conditions than what is presented in the SAR image. This is one of the biggest sources of uncertainty in ice charts - when analysts have to guess the situation only from SAR features. An example of such case was discussed in Gegiuc et al. (2018).**

**It is true that the sample size is small. The main reasons for this were the limited resources: limited time of ice analysts for this experiment and the limitations related to use of eye tracking hardware and software, they needed to be**

**monitored by a scientist familiar with the setup during the analyses. Despite of the limited data set we gained informa-**

100    **tion and preliminary statistics on the eye fixations during the ice analysis and their relation to the local details in SAR imagery and ice conditions. This information can also be used to design better experiments and design of collection of larger data sets. Also some of the collected information could possible be utilized in automated algorithm development.**

105    Last, this method is not easily feasible, economically or timewise, to use with ice analysts. Though the cost of the eye-tracking software is a factor, the usefulness is more related to the amount of time ice analysts are able to spare outside of operations to provide feedback towards these types of intercomparison studies. This approach is far more cumbersome to implement and open to further interpretation, rather than developing a more scientific metric-based evaluation to analyze uncertainties with subjectivity in ice charts.

110    **AR: The main idea was not to make comparisons between ice analysts and the uncertainties (deviation of the analyses of several ice analysts) but to test how the visual interpretation of SAR imagery is performed in general in ice analysis and if there is any relation to the local SAR statistics and features (edges etc) and ice conditions.**

The current method does not support the outcome that "the long fixation duration are connected with larger uncertainties

115    in the final ice charts" stated on P27 L5, as there are a number of other factors which can be affecting the analysts decision-making. It is important that these types of studies are being developed so we can  understand the human bias in ice charts and it is great to see these new and innovative approaches. However, the experiment in this methodology needs to be 1) redesigned to allow the analysts to include additional sources of information that they would regularly require for routine ice analysis, as well as 2) putting them in their normal working environment using the common systems that they are familiar with. This will allow

120    them to use all necessary sources of information without compromising the functionality or spatial resolution in which they're familiar and will allow for more appropriate assessments on the subjective nature  between expert and non-expert analysts.

**AR: We agree with RC1 that the method presented here is not suitable for operational ice charting. However, the use of eye tracking in operational ice charting context would be possible in a more automated way, where the collection and processing of the eye data would be somehow integrated or synchronized with the sea ice charting software. We hope**

125    **that by showing that eye movement data is useful in the context of ice charting, than we can obtain better funding to conduct more in-deep studies and eventually to integrate eye trackers in the sea ice services. We believe that novice sea ice analysts would greatly benefit during their learning and practice of ice charting, as well as the more expert analysts which could provide faster feedback to the student. Lastly, the International Ice Charting Working Group has reported that one of the sources of uncertainty in ice charting is related to human skill, data available for analysis. These are**

130    **the points we tried to study here. To involve the eye tracking device in the actual ice charting process would not be an easy task. It will require a lot of work and adjustment to perform without disturbing the actual ice charting process. However, if we want to collect a comprehensive data set this should be done in a controlled way in the future.**

Reference: Cheng, A., Casati, B., Tivy, A., Zagon, T., Lemieux, J. F., Tremblay, L. B. (2020). Accuracy and inter-analyst agreement of visually estimated sea ice concentrations in Canadian Ice Service ice charts using single-polarization RADARSAT-2. The Cryosphere, 14(4), 1289-1310.

**2 Specific comments by RC1:**

**We will address the specific issues in the revised version of the manuscript.**

The following are specific comments from the first part of the paper:

P2 L9: use of term inconsistencies

P2 L10 Replace "miss-classification" with "misclassification".

P2 L10-12: wouldn't areas that require more cognitive effort be prone to less miss-classification? The following sentence then states that areas less restrictive to navigation are more flawed. Ice analysts spend more time on areas where high traffic areas are known to be, including areas that are more restrictive, as a safety precaution. If areas are less restrictive, ideally they would require less cognitive effort. These sentences contradict one another. Additionally, the combination of both sea ice regimes and level of regulation in a given area for ice charting has significant implications on how analysts focus on the attention to detail in a particular area. Sea ice operations in the Baltic and the Arctic are often confused so this should specify that this paper is focused on the Baltic.

P2 L12-13: What is not being highlighted is that experts are able to map large sea ice covered areas because they have continuity in observing how the ice is changing on a daily or weekly basis. This is very different from someone who understands how to interpret sea ice in SAR imagery and may be looking at it for the first time, without having knowledge of environmental conditions in the area. This statement is incredibly misleading.

P2 L15: What is the purpose of this paper? To use eye-tracking as a metric to calculate uncertainty? If so, this should be stated clearly.

P2 L14 Confusion of terminologies, "open ice" and "very open ice" refer to concentration and not to whether the ice thicknesses are mixed.

P2 L16: What is meant by "large areas?" Does this mean synoptic? If so, up-to-date information is required at meter scale resolution, especially for tactical navigation. For route planning, large-scale information is more useful. Depending on the area, navigators require both but the "typically over large areas" simplifies the needs of maritime users and their data requirement needs.

P2 L24: need to include the challenges that snow cover and melt have on the surface roughness because this is the key challenge in sea ice monitoring and one of the main reasons for ice charts continuing to be fully manual, as opposed to semi-automated.

P2 L26: Sentence needs to be revised.

P2 L31: Is there a metric used in this comparison?

165    P4 L7-8 MANICE gives only a brief outline of ice charting practices, specific to Canadian Ice Service, and more the type of information content to be found in ice charts.

P4 Table 1: New ice and level ice categories are typically not used in sea ice concentration analysis. P4 L16-18 Check Zakhvatkina et al 2019 reference is an overview, maybe more just what AARI have been doing?

P5 L1: Omit "Even"

170    P5 L10: Omit "for long"

P6 L1: Revise, awkward. Suggestion: "The FIS ice analysts have experience with analysing SAR images for drawing sea ice charts since....."

P6 L5: This does not need to be a separate sentence and Table 3 can just be referenced at the end of sentence from P6 L4.

P6 L7: Does this refer to Table 3 or Figure 3?

175    Pg6 L10: Specify original resolution

P6 L9-10 Specify the original resolution of RADARSAT-2 ScanSAR Wide. Depending on the processing it can be 100 or 50 m.

P7 L18 "an external monitor with a 22" diagonal size, similar to the ones used in the operational ice charting". FMI typically uses a Wacom digitizing screen so that the analyst is looking directly at and drawing on the image being processed. Was this

180    set up changed for this experiment?

P7 L27-28 "the SAR images were opened and viewed with an image viewing program (Irfan View)" This is again different from the ArcGIS software used by FIS ice analysts.Pg10 L10: Replace "fore" with "for a"

P10 L28: Who does "he" refer to? E1 or E2? Probably the use of pronouns should be neutral throughout the paper to maintain neutrality in subjects.

185

Gegiuc, A., Similä, M., Karvonen, J., Lensu, M., Mäkynen, M., and Vainio, J.: Estimation of degree of sea ice ridging based on dual-polarized C-band SAR data, The Cryosphere, 12, 343-364, https://doi.org/10.5194/tc-12-343-2018, 2018.

---

## Author Comment (AC2)

**Author Responses to RC2: Comment on tc-2022-8", Anonymous Referee 2, 19 Apr 2022**

Citation: https://doi.org/10.5194/tc-2022-8-RC2

The Author Responses are marked with bold text.

**1    General comments by RC2**

Review of: Visual Interpretation of Synthetic Aperture Radar Sea Ice Imagery by Expert and Novice Analysts: An Eye Tracking Study. Alexandru Gegiuc et al.

The paper presents the results of an experiment in the use eye tracking technology to "identify elements behind uncertainties typically introduced during the process of sea ice charting using satellite synthetic aperture radar (SAR) imagery" by comparing the efforts of experts to novices.

Unfortunately – almost all of the papers insights into the problem could have been written down at the outset. The paper demonstrates the obvious – that SAR images of sea ice are complex and difficult to interpret particularly in areas that contain the signatures of several different phenomena. And that even expert analysis can take different approaches and produces inconsistent results. In fact the Novice analysis is almost completely superfluous – you could easily delete it from the paper focus on the expert analysis (which you do for pages 13 to 22 anyway) and which are the basis for almost all the conclusions (except that Novices are bad at interpreting SAR sea ice imagery). .

**A.R.: We will shorten the novice analysis part but still see it compared to expert analysis as an important piece of information. For example in considering automated SAR image algorithm development it is important to know how the experienced ice analysts perform and to avoid novice behavior.**

The expert analysis is used to conclude that "eye movement data in further studies to deepen this kind of knowledge and to understand the uncertainties introduced" [in ice charting]. And while that seems to be a reasonable and noteworthy result-there seems to be no path by which these results can be effectively translated into practice ("link between the observer and the automated method has not yet been established"). Analysts looking at imagery is slow, inefficient and (as the authors report – inconsistent) and is not a viable way to process the ever increasing amount of available SAR sea ice data. It would seem that establishing this link is one of the most important thing to do. (rather than compare expert and novice with the obvious conclusion)

It was frustrating that after reading through 26 pages – with lots of detail on pages 11 to 24 only for the authors decide to tell me on page 26 that the "main findings are qualitative in nature" (so why all the detail) and "more SAR data and more

ice analysts are needed for a comprehensive study". Really - it would have been nice to know all that up front (I would have skipped the details).

30 **A.R.: We will improve the structure in the revised version and try to focus on the most relevant issues.**

The paper in fact immediately raised a red flag when I discovered it relied on imagery from 2010 to 2012 – i.e. imagery 10 to 12 years old as its primary focus. The study area in the Baltic Sea has approximately 2000 passes from the Sentinel-1 spacecraft between 2014 and 2022. It is particularly odd to focus on old imagery given the references on page 3 to... "The

35 basis for the Baltic Sea ice charting at FIS is daily SAR mosaic.... The mosaic is updated once per day, typically in the morning, to include most recent available SAR" So why was RS2 and not S1 used for the study?

**A.R.: It is not relevant which data sets are used. The RS-2 and S-1 C-band SAR imagery are very similar. For this study some images with different and interesting ice conditions were and easy access selected. They just happened to be older data. We do have a lot of SAR data but there were other reasons limiting the amount of data to be used (limited**

40 **time of ice analysts and the setup and maintenance of the eye tracking system during the experiments). This has also been a learning process for us and we hope that we'll be able to set up a more sophisticated eye tracking experiment connected to the actual ice charting process to collect more representative data sets. However, this will require significant additional resources (human resources and funding for hardware and salaries).**

45 Please go back and decide what paper you want to write Novice v Expert SAR Ice Analysis (nothing to see here / expected results) or What Can the Eye Tracking of Experts Tell Us about How to Improve Sea Ice Charting (potentially really useful) – in particular please tie it to a potential way to quantify error in charting or improve ice classification. But certainly don't spend pages on details and then tell me at the end they don't matter. (and that you need a bigger study to get any useful answers)

**A.R.: The main idea is related to collecting information on how the ice analysis is performed by an experienced ice**

50 **analyst and to gain information to possibly improve ice charting and automated sea ice SAR analysis. We'll also try to improve the comparison between novices and experienced ice analyst to identify where the experienced analysts behave differently. This information is very valuable information for developing automated algorithms and also to be able to guide the novices to perform better.**

**2 Specific comments by RC2:**

55 Specific Issues:

**A.R.: We will address the specific issues in the revised version of the manuscript.**

Abstract: "We also show that the experts are able to correctly map large sea ice covered areas only by looking at the SAR images" I am not quite sure how they reached this conclusion give that the Balitc Sea is a rather small mostly enclosed marginal sea.

60     Section 2.1 "We used five RADARSAT-2 (RS-2) ScanSAR Wide images covering different regions of Baltic sea across three different winter seasons". Why was it important to use 3 seasons – all imagery from February. The sea ice conditions might or might not be similar.. (or was that the point?)

    "In FIS, the original SAR images are typically reduced in size for easier manipulation and saving disk space. This reduces the amount of detail available for analysis. The 100 m resolution of the SAR imagery used by the FIS analysts is lower than the

65 original resolution. Here we used the same down-scaled resolution for the RS-2 SAR images" This is one of those paragraphs that make me wonder if this paper has been sitting on a shelf for a decade and someone just dusted it off. Are the authors really having disk space limitation issues? A ScanSAr tiff is about 300 MB. They never specify the original resolution (its 50 m for ScanSAR – meaning at most you saved at most factor of 4 by moving to 100 m). What other preparation of the imagery were undertaken? Are they calibrated to produce Normalized Radar Cross Section so that the SAR signatures across years can be

70 properly inter-compared or was it just the digital number?

    Section 2.2 "Two novices (N1 and N2) with little or no familiarity with classification of SAR sea ice imagery participated to the study" and Section 3.1 "We instructed the participants to look at the selected images and interpret the content verbally. . . .When looking at the SAR images, the participants had the task to describe their content freely by identifying sea ice types and features and classify them as they would in a typical ice charting routine"

75     So this would appear to be contradictory – The authors expected the Novices to verbally describe sea ice types and features – when they had "little or no familiarity with classification of SAR sea ice"

    Section 3.1: "while the SAR images were opened and viewed with an image viewing program (Irfan View) which allowed users to freely change the scale or pan the viewed images." Why this program – were the participants familiar with it? Was the SAR imagery presented in isolation? I.E was there a map reference for location? Reference scale for NRCS gray scale

80 values or a map scale to indicate the size of the image and features?

    Figure 3: Why are the land areas presented as both white and black. Is this display representative of how the images were displayed to the participants – without scales or map references?

    Section 3.3 "We divided the gaze data into segments that correspond to the scanning phase and the analysis phase" Is this a standard practice when performing eye tracking work?

85     Section 3.3 "We computed the average dwell time, fixation duration mean, standard deviation, and fixation density (number of fixations per ice area)." Is this standard reporting metrics when performing eye tracking work? If so what kinds of information do these metrics convey?

    Section 4. The title is Visual Interpretation of Synthetic Aperture Radar Sea Ice Imagery by Expert and Novice Analysts. Yet section 4 is devoted to the experts and there appears to be no corresponding section for the Novices. As stated above – the

90 authors could delete the Novice analysis entirely and not lose much)

    Section 4.1 "The first difference between experts (E) and novices (N) was noticed right away, based on their reactions when they were shown the first SAR image." . . . and the first difference is associated with. . . . . . .? They never explicitly call out a second (subsequent) difference..

Section 4.1 "Even if novices recognized fast (in the first five seconds) that the image shown is a SAR image" Even if!!!?

Did either novice need more than 5 seconds to determine they were looking at a SAR image given that the other images were of easy to recognize natural images (human face, a flower, a fish, a cat and a bird) displayed on entirely different system (Tobii Studio vs Irfan View) - Seriously

Section 4.2.2: "This is an interesting result, showing that the experts need only few seconds (5-14) of fixation time to be able to classify an ice area. Also it underlines a difference in style of analysis. E2 spent in average about five times longer analyzing an image and about three times longer analyzing an ice area than E1. These differences could be explained by the inequality between the two experts in terms of years of experience and of training." I'm not sure its really all that interesting. Is there an alternative explanation? I don't not believe that the years of experience (10 v 25) can account for that amount of difference. After 10 years of experience a good analyst has seen pretty much everything. More likely this difference is explained by "personal styles of analysis" and not expertise.

Section 5: Limitations and Future work "The aim of this study was to act as a proof of concept study... The sample number in our study is low, and thus, the main findings are qualitative in nature.." All of this should have been stated at the beginning of the paper. Citation: https://doi.org/10.5194/tc-2022-8-RC2